# Changes in Soil Rhizobia Diversity and Their Effects on the Symbiotic Efficiency of Soybean Intercropped with Maize

Zeyu Cheng [1,†], Lingbo Meng [2,†], Tengjiao Yin [1], Ying Li [1], Yuhang Zhang [1] and Shumin Li [1,*]

1   Resource and Environmental College, Northeast Agricultural University, Harbin 150030, China
2   School of Geography and Tourism, Harbin University, Harbin 150086, China
*   Correspondence: shuminli@neau.edu.cn
†   These authors contributed to this work equally.

**Abstract:** It has been established that maize/soybean intercropping can improve nitrogen use efficiency. However, few studies have addressed how maize/soybean intercropping affects nitrogen-fixing bacterial diversity and N fixation efficiency of intercropped soybean. In this study, nitrogen-fixing bacterial communities, N fixation efficiency, and their relationships with soil properties under three nitrogen fertilization application rates (N0 0 kg/ha, N1 40 kg/ha, N2 80 kg/ha) were explored through field experiments. Nitrogen fixation and nitrogen-fixing bacteria diversity were assessed using $^{15}N$ natural abundance, Illumina high-throughput sequencing, and *nifH* (nitrogen fixation) gene copies quantification in the rhizosphere soil of intercropped soybean. The results showed that nitrogen application rates significantly decreased the nitrogen-fixing bacteria diversity, nitrogen fixation efficiency, and *nifH* gene copies in the rhizosphere soil. Nitrogen fixation efficiency, nodule number, and dry weight of intercropped soybean were highest in the N0 treatment, and nitrogen fixation was the highest in the N1 treatment. The nitrogen-fixing efficiency in N0, N1, and N2 treatments increased by 69%, 59%, and 42% and the nodule number of soybean was 10%, 22%, and 21%, respectively, compared with monocultures. The soybean nitrogen-fixing bacteria diversity in intercropping under N0 and N1 treatments significantly increased compared with monocultures. There was a significant positive correlation between soil *nifH* gene copies and N fixation efficiency and a negative correlation with soil available nitrogen. *Bradyrhizobium* abundance in soybean rhizosphere soil decreased significantly with the increase in nitrogen application rates and was significantly correlated with soil AN (available nitrogen) and pH content in the soybean rhizosphere. These results help us to understand the mechanisms by which nitrogen use efficiency was improved, and nitrogen fertilizer could be reduced in legume/Gramineae intercropping, which is important to improve the sustainability of agricultural production.

**Keywords:** nitrogen-fixing efficiency; nitrogen-fixing bacteria; intercropped soybean; $^{15}N$ natural abundance; *Bradyrhizobium*

## 1. Introduction

Intercropping systems are among the important cultural practices to further increase yield in China's agriculture. Selecting the appropriate crop intercropped, yield [1], and nutrient use efficiency [2] have more significant advantages than the corresponding monocropping. Many legumes/gramineous intercropping combinations, such as wheat/soybean [3], cowpea/corn [4], lentil/barley, etc., are common beneficial combination systems. Maize/soybean intercropping is a frequently used intercropping planting pattern and has been found to have certain advantages in nutrient utilization, significantly increasing the nitrogen equivalent ratio [5]. At the same time, maize/soybean intercropping can not only significantly promote rhizosphere microbial abundance but also affect rhizosphere microbial diversity [6,7]. Some studies showed that Gramineae could absorb nitrogen fixed by intercropped legumes and, at the same time, secrete root exudates to influence legumes' N

fixation [8–10]. Except that, in the legume/cereal intercropping system, due to the large nitrogen amount absorbed by the cereal crops, soil mineral nitrogen is maintained at a relatively low level, reducing the nitrogen-induced inhibition of legume nitrogen fixation, which can promote legume nitrogen fixation from the air to satisfy their own growth [11]. In addition, due to the ecological niche differences [12], the two intercropping crops have different plant heights and alternate planting rows, which reduces the competitive effect. Moreover, the different root depths of the two crops lead to growth at different soil levels, reducing their competition. These would also lead to differences in nitrogen fixation efficiency in intercropping compared to monoculture. Studies have shown that intercropping significantly increased crop yield and promoted nodulation and nitrogen fixation of legumes [13]. The nitrogen application levels and interspecific interaction in intercropping systems affected the complementary utilization of nutrients, growth of nodules, and nitrogen fixation of legumes. Therefore, the advantages of intercropping partially depend on below-ground interspecific plant interactions [14], which include interspecific facilitation, such as the complementary utilization of N resources [10] and niche differences [15].

In general, legumes fix air nitrogen by forming a symbiotic system between nitrogen-fixing soil microorganisms and legume roots. Plants can only absorb free nitrogen in the atmosphere under the action of nitrogen-fixing microorganisms. Nitrogen-fixing bacteria are among the bacteria involved in nitrogen fixing. Rhizosphere nitrogen bacterial diversity and structure will influence the nitrogen fixation of legumes in intercropping. Different planting cultivation would affect soil microbial community structure, microbial quantity, and microbial activity [16]. Therefore, it is necessary to study the changes of nitrogen-fixing bacteria in the rhizosphere to verify the mechanism of high N fixation efficiency in the maize/soybean intercropping system. It has been shown that the intercropping of legumes and grasses could significantly increase the abundance of rhizosphere soil microorganisms [17,18]. Nitrogen-fixing bacteria regulate the nitrogen-fixing process through the 70K Da nitrogen-fixing enzyme encoded by the *nifH* (nitrogen fixation) gene [19]. Therefore, exploring the changes in the *nifH* gene number in the rhizosphere soil is helpful in clarifying the influence on legume fixation efficiency and development.

Nitrogen-fixing bacteria are important in the soil nitrogen cycle [20] and are affected by soil nitrogen levels. The change in Azotobacter diversity is beneficial to plant growth [21]. A certain amount of nitrogen fertilizer would improve the nitrogen-fixing efficiency of nitrogen-fixing bacteria in soybean [22]. Nitrogen application levels have an important effect on rhizosphere microorganisms and community growth and development changes. Some studies have found that soil available nitrogen has a significant effect on bacterial community composition [23], and short-term fertilization also changes the community composition of rhizobia [24]. Fertilization increased the abundance of nitrogen-fixing bacteria species and was beneficial to increase the number and variety of nitrogen-fixing bacterial genes in the black soil area of Northeast China [25]. Evaluation of rhizosphere *nifH* gene diversity in two sorghum (*Sorghum Moench*) varieties suggested that nitrogen application was the main factor affecting the nitrogen-fixing bacteria community structure in sorghum [26]. However, long-term fertilization would greatly inhibit bacterial nitrogen fixation; other studies showed that the increase in nitrogen fertilizer inhibited the expression of the *nifH* gene [5]. Nitrogen-rich soil inhibited soybean symbiosis and nitrogen fixation, which resulted in decreased soybean nodule quantity and size and decreased nitrogen fixation activity [27]. Planting methods and soil nutrients are important factors determining rhizosphere microbial diversity and community composition [28]. However, there is a limited number of studies about the community structure of nitrogen-fixing bacteria and *nifH* gene abundance in the maize/soybean intercropping under different N application rates.

Therefore, in the present experiment, three nitrogen application levels were evaluated in the maize/soybean intercropping system. The $^{15}$N natural abundance method, high-throughput sequencing technology, and real-time fluorescent quantitative PCR were used to study the community structure of nitrogen-fixing bacteria and *nifH* gene abundance in

the soybean rhizosphere. The aims of this study were: (i) to quantify the nitrogen fixation efficiency of intercropped soybean as affected by N application rates. (ii) to evaluate the changes of nitrogen-fixing bacteria community diversity and abundance in the rhizosphere of intercropped soybean; (iii) to reveal the relationship between *nifH* gene abundance and nitrogen fixation efficiency of intercropped soybean. These results will further explain why N utilization efficiency is improved in the maize/soybean intercropping system, which is valuable for reducing N use in agricultural ecological systems.

## 2. Materials and Methods

### 2.1. Experimental Design

A field experiment was conducted at the experimental station of Northeast Agricultural University, Acheng District, Heilongjiang Province, China. The station is in the temperate continental monsoon climate, with an annual precipitation of 530 mm and annual accumulated temperature of 2800 degrees Celsius. The soil type was black soil, which is mollisol according to the USDA soil classification. The basic physical and chemical properties of the soil were as follows: soil OM (organic matter): 29.6 g/kg; TN (total N): 1.45 g/kg; (AP) available P: (Olsen P) 28.9 mg/kg; AK (available K): 122.52 mg/kg; AN (available nitrogen): 126 mg/kg; pH: 6.1 (1:2.5 *w/v* water).

The experiment was conducted in a two-factor split-plot design. The main factor was nitrogen application, with the three nitrogen application rates (N0, N1, and N2) in monoculture soybean and maize/soybean intercropping systems, respectively. The N0, N1, and N2 application rates were 0 kg N/ha, 40 kg N/ha, and 80 kg N/ha, respectively. The planting patterns were the secondary factor w, namely soybean monocropping (SS) and maize/soybean intercropping (IS). In the monoculture plots, soybean was grown throughout the eight rows. Soybean plant spacing was 8.5 cm, and planting density was 196,000/ha. Maize row spacing was 60 cm, and plant spacing was 17 cm. Urea was used as a nitrogen fertilizer for each treatment, with the same application amount in intercropping and monoculture. The basic fertilizer was applied with soybean per the standard treatment guidelines, and the application amount was 0 kg N/ha, 40 kg N/ha, and 80 kg N/ha, respectively. Superphosphate and potassium sulfate was used as the phosphorous potassium fertilizer 120 kg $P_2O_5$/ha was supplied as triple superphosphate, and 100 kg $K_2O$/ha was supplied as potassium sulfate. The experiment had 6 treatments with 3 replications and a total of 18 plots. Each plot's dimensions were 4.8 m (width) $\times$ 10 m (length) and included eight rows with a row spacing of 0.6 m. The maize variety "Xianyu 335" and soybean variety "Dongnong 252" were planted. The sowing time was 2 May 2019, and the harvest time was 1 October 2019.

### 2.2. Plant Sampling and Analysis

Five soybean plants were taken at random in the sampling area at the mature stages of the crop. After the harvest, plants were taken back to the laboratory, and the stalks and grains were separated, dried in an oven at 105 °C for 30 min, and then dried at 80 °C to a constant weight. The samples were ground thoroughly for the $^{15}$N abundance determination and N concentration analysis.

The samples were digested with $H_2SO_4$-$H_2O_2$, and the N concentration was measured by the Kjeldahl method. The $^{15}$N abundance was measured with a stable isotope mass spectrometer (ISOPRIME).

### 2.3. Analysis of Soil Physicochemical Characteristics

The Kjeldahl distillation method [29] was used for the total nitrogen (TN) determination. The available P (AP) was extracted with a 0.5 mol/L $NaHCO_3$ solution and then measured using the molybdenum antimony-D-isoascorbic acid colorimetry (MADAC) method [30]. The available potassium (AK) was determined by flame photometry using a 1 mol/L $NH_4OAc$ neutral extraction. The available nitrogen (AN) was determined by the diffusion absorption method [31]. The soil pH was measured in a 1:2.5 soil–water

suspension using a glass electrode. The soil organic matter (OM) was determined by the method of soil digestion with hot acid dichromate [32].

*2.4. Measurement of Soybean Nodules and Nodules Dry Weight*

Soybean root nodules were collected at the pod setting stage. Five soybean plants from each treatment were randomly selected for the determination. The roots of soybean plants were excavated from the field (20 cm(long) $\times$ 20 cm(wide) $\times$ 25 cm(deep)), then placed in sealed bags and taken then back to the laboratory. Roots were shaken off the excess loose soil on a kraft paper. The soil was collected and sieved to wash off the soil and pick out the nodules. The soil attached to the roots was brushed off and passed through a 2 mm sieve. The soil samples were then placed into a centrifuge tube and stored in a $-80$ °C refrigerator for the analysis of nitrogen-fixing bacteria diversity. The nodules attached to the roots and the sieves were counted. Then the nodules were dried at 80 °C to a constant weight, and their dry weight was taken.

*2.5. Determination of Nitrogen-Fixing Bacteria Diversity*
2.5.1. DNA Extraction and PCR Amplification

Total microbial genomic DNA was extracted from rhizosphere soil samples using the E.Z.N.A.® soil DNA Kit (Omega Bio-tek, Norcross, GA, USA) according to the manufacturer's instructions. The quality and concentration of DNA were determined by 1.0% agarose gel electrophoresis and a NanoDrop® ND-2000 spectrophotometer (Thermo Scientific Inc., Waltham, MA, USA) and were kept at $-80$ °C prior to further use. The *nifH-F_nifH-R* variable region was amplified by PCR with *nifH-F* (AAAGGYGGWATCG-GYAARTCCACCAC) and *nifH-R* (TTGTTSGCSGCRTACATSGCCATCAT) [33,34] primers and sequenced using the Illumina Miseq platform. The PCR reaction mixture including 4 μL 5 $\times$ Fast Pfu buffer, 2 μL 2.5 mM dNTPs, 0.8 μL Forward Primer (5 μM), 0.8 μL Reverse Primer (5 μM), 0.4 μL Fast Pfu polymerase, 0.2 μlBSA,10 ng of template DNA, and ddH$_2$O to a final volume of 20 μL. PCR amplification cycling conditions were as follows: initial denaturation at 95 °C for 3 min, followed by 35 cycles of denaturing at 95 °C for 30 s, annealing at 55 °C for 30 s and extension at 72 °C for 45 s, and single extension at 72 °C for 10 min, and end at 10 °C. All samples were amplified in triplicate. The PCR product was extracted from 2% agarose gel and purified using the AxyPrep DNA Gel Extraction Kit (Axygen Biosciences, Union City, CA, USA) according to the manufacturer's instructions and was quantified using Quantus™ Fluorometer (Promega, Madison, WI, USA).

2.5.2. Illumina MiSeq Sequencing

Purified amplicons were pooled in equimolar amounts and paired-end sequenced on an Illumina MiSeq PE300 platform (Illumina, San Diego, CA, USA) according to the standard protocols by Majorbio Bio-Pharm Technology Co. Ltd. (Shanghai, China). The raw sequencing reads were deposited into the NCBI Sequence Read Archive (SRA) database (Accession Number: PRJNA945238).

2.5.3. Data Processing

Raw FASTQ files were de-multiplexed using an in-house perl script and then quality-filtered by fastp version 0.19.6 and merged by FLASH version 1.2.7 with the following criteria:

(i) the 300 bp reads were truncated at any site receiving an average quality score of <20 over a 50 bp sliding window, and the truncated reads shorter than 50 bp were discarded; reads containing ambiguous characters were also discarded; (ii) only overlapping sequences longer than 10 bp were assembled according to their overlapped sequence. The maximum mismatch ratio of the overlap region is 0.2. Reads that could not be assembled were discarded; (iii) samples were distinguished according to the barcode and primers, and the sequence direction was adjusted, exact barcode matching, and 2 nucleotide mismatches in primer matching. The bacterial composition and differences between groups were analyzed

and compared on the Majorbio I-Sanger cloud platform and in the *nifH* database. The RDP classifier Bayes Algorithm was used to perform taxonomic analysis on 97% similar OTU representative sequences, and statistics at each classification level in each sample community were performed (http://sourceforge.net/projects/rdp-classifier/ accessed on 12 December 2019). This part of the sequencing process was completed by Shanghai Majorbio Bio-pharm Technology Co., Ltd., (Shanghai, China).

### 2.6. Determination of nifH Gene

ABI 7500 fluorescence quantitative PCR instrument (USA) and SYBRGreen Real-Time PCR kit (ABI Power SybrGreen qPCR Master Mix(2X)) were used. The copy number of bacterial nifH gene was quantitatively analyzed by the SYBRGreen I method. The primers were *nifH-F* (AAAGGYGGWATCGGYAARTCCACCAC) and *nifH-R* (TTGTTSGCSGCRTA-CATSGCCATCAT) [33,34].

The conversion formula of gene copy number is (copies/$\mu$L) = concentration (ng/$\mu$L) $\times 10^{-9} \times 6.02 \times 10^{23}$/(molecular weight $\times$ 660).

The copy number was calibrated using CT = $-$k lg X0 + b. The number of qpcr was repeated 3 times, which was normalized by internal reference genes.

Preparation of the standard curve: ten times gradient dilution of each constructed plasmid; 45 $\mu$L diluent + 5 $\mu$L plasmid; generally conduct 4–6 points, through the preliminary experiment, respectively, and select $10^{-3} \sim 10^{-8}$ diluent of standard for the preparation of the standard curve.

### 2.7. Method of Calculating Soybean Nitrogen Fixation Efficiency

After the samples were harvested, $\delta^{15}$N data were obtained by the elemental analyser-isotope ratio mass spectrometer, and then were calculated:

$$\delta^{15}N = [\text{atom\% } ^{15}N_{(\text{sample})} - \text{atom\% } ^{15}N_{(\text{standard})}]/\text{atom\% } ^{15}N_{(\text{standard})} \times 1000$$

$\delta^{15}$N is the difference between $^{15}$N of the sample and $^{15}$N of the atmosphere, atom% $^{15}$N $_{(\text{sample})}$ is the $^{15}$N atomic abundance of the sample, and atom% $^{15}$N$_{(\text{standard})}$ is the atmospheric standard $^{15}$N atomic abundance (0.3663%).

Since grain and straw were treated separately, the above formula $\delta^{15}$N was calculated by the weighted average of the two $^{15}$N.

$$\delta^{15}N_{\text{maize}} = [N_{\text{maize grain}} \times \delta^{15}N_{\text{maize grain}} + N_{\text{maize stover}} \times \delta^{15}N_{\text{maize stover}}]/(N_{\text{maize grain}} + N_{\text{maize stover}})$$

$$\delta^{15}N_{\text{soybean}} = [N_{\text{soybean grain}} \times \delta^{15}N_{\text{soybean grain}} + N_{\text{soybean stover}} \times \delta^{15}N_{\text{soybean stover}}]/(N_{\text{soybean grain}} + N_{\text{soybean stover}})$$

In the above formula, $\delta^{15}N_{\text{maize grain}}$ represents $\delta^{15}$N of maize grain, $\delta^{15}N_{\text{maize stover}}$ represents $\delta^{15}$N of maize stover, $\delta^{15}N_{\text{soybean grain}}$ represents $\delta^{15}$N of soybean grain, and $\delta^{15}$N $_{\text{soybean stover}}$ represents $\delta^{15}$N of soybean stalk.

$$\%\text{Ndfa} = (\delta^{15}N_{\text{maize}} - \delta^{15}N_{\text{soybean}})/(\delta^{15}N_{\text{maize}} - B) \times 100$$

The B value corresponds to the $\delta^{15}$N value of the nitrogen-fixing soybean plants with no nitrogen supplied but with all other nutrient requirements supplemented. Thus, it considers legume plants that use atmospheric nitrogen as the only source of nitrogen.

Ndfa = %Ndfa $\times$ (Nsoybean grain + Nsoybean straw) corresponds to the nitrogen fixing of nitrogen-fixing bacteria in soybean from the air.

Nitrogen uptake = (Nsoybean grain + Nsoybean straw) $-$ Ndfa [35].

### 2.8. Statistical Analysis

The statistical analysis was carried out by R language (R, V4.2.1); that is, the difference of different treatments was analyzed by one-way variance analysis (ANOVA), and the significance of the difference by Duncan multiple repetition range test ($p < 0.05$); the

difference of two-ways ANOVA variance analyzed the differences among different factors. The R language "fBasics package" checks the normal distribution; "ggplot2 package" was used for data visualization and analysis; "vegan package" for RDA analysis; "ggcor package" for Mantel Test analysis; "cancor package" for Person correlation analysis.

## 3. Results

### 3.1. Soybean Nitrogen Fixation Efficiency, Biological N Fixation and N Uptake from the Soil

The nitrogen-fixing efficiency of nitrogen-fixing bacteria in soybean was significantly affected by nitrogen application rates ($p < 0.01$) and cultivation patterns ($p < 0.01$). The soybean nitrogen fixation efficiency decreased as nitrogen application rates increased, with rates of 69%, 59%, and 42% measured in the N0, N1, and N2 treatments, respectively. In contrast, intercropped soybean showed significantly higher nitrogen fixation efficiency compared to monocropped soybean, with increases of 20%, 21%, and 10% observed compared to monoculture, respectively (Figure 1a).

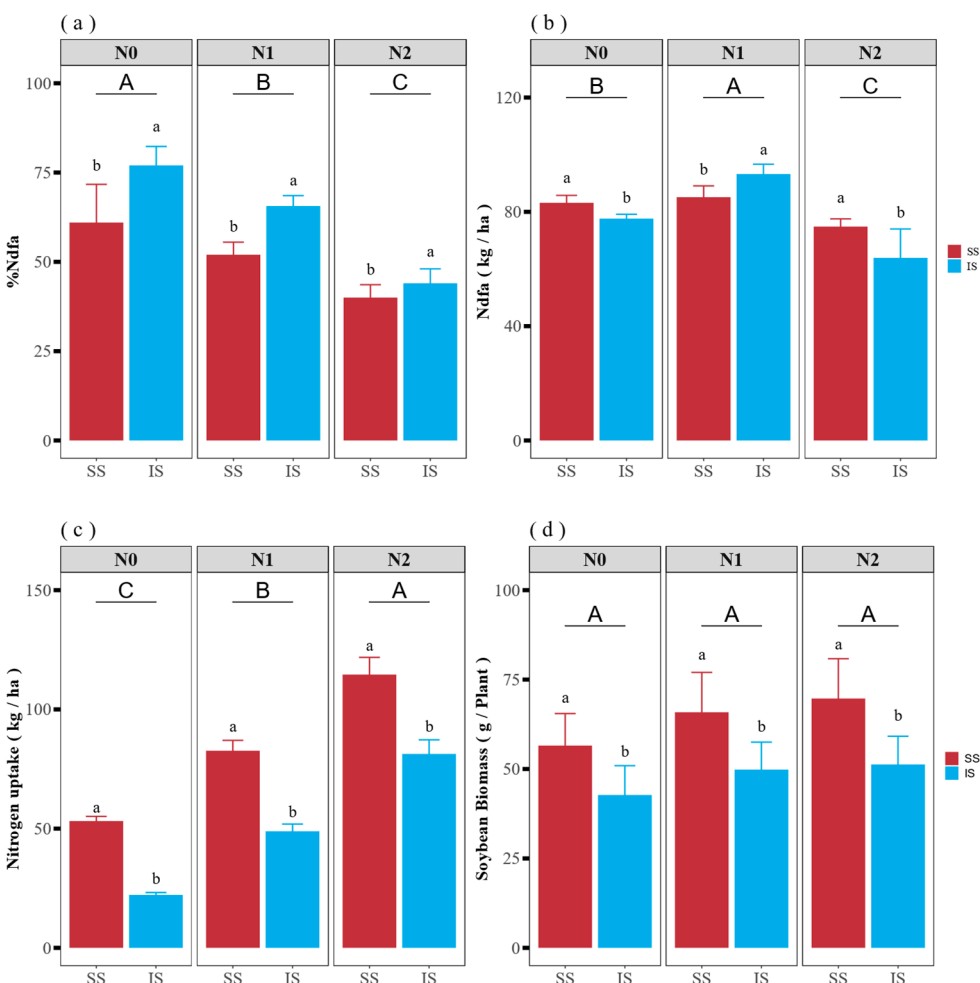

**Figure 1.** Nitrogen fixation efficiency (**a**) and biological nitrogen fixation (**b**) and nitrogen uptake in the soil (**c**) and biomass (**d**) of soybeans under different nitrogen application levels. Different lowercase letters indicate significant differences between single intercropping at the same nitrogen supply level ($p < 0.05$), and different capital letters indicate significant differences between the same nitrogen supply level ($p < 0.05$). Data are presented as a Mean ± SD ($n = 5$).

The biological nitrogen fixing of nitrogen-fixing bacteria in soybean was determined by the $^{15}$N method. The soybean biological nitrogen fixation was significantly influenced by nitrogen levels ($p < 0.01$) and cultivation patterns ($p < 0.01$) and was 80 kg/ha, 91 kg/ha, and 70 kg/ha (average) in the N0, N1, and N2 treatments, respectively. However, the

biological nitrogen fixation in the N1 treatment was significantly higher compared to other treatments and was increased by 4% compared to monocropping. On the other hand, biological nitrogen fixation in monocropping was 8% and 16% higher compared to intercropping in the N0 and N2 treatments (Figure 1b). In contrast, soil nitrogen uptake by soybean showed an opposite response to that of nitrogen fixation efficiency with regard to intercropping (Figure 1c). Nitrogen uptake from the soil increased with nitrogen application rates ($p < 0.01$). However, in the soybean–maize intercropping system, nitrogen uptake from the soil decreased by 57%, 41%, and 32% in N0, N1, and N2 treatments, respectively, compared to the monoculture ($p < 0.01$).

The soybean biomass showed an increasing trend with the increase of nitrogen application rate, but the effect of nitrogen application rate on the soybean biomass did not reach a significant level (Figure 1d). On the other hand, the soybean biomass was significantly influenced by different cultivation methods, and it was significantly higher under monoculture than intercropping. Specifically, the soybean biomass under monoculture increased by 32.35%, 32.20%, and 35.98% compared with intercropping, respectively, at N0, N1, and N2 levels.

### 3.2. Soybean Nodules Numbers and Nodules Dry Weight

Nodule formation is a crucial factor for symbiotic nitrogen fixation and a key indicator for studying biological nitrogen fixation. The number of nodules decreased significantly with increasing nitrogen application rates. However, when soybean was intercropped with maize, the number of nodules increased by 10%, 22%, and 21% in the N0, N1, and N2 treatments compared to the monoculture (Table 1, $p < 0.01$). No significant differences were observed in nodule dry weight among different treatments. On average, the soybean nodules' dry weight in the N2 treatment was lower than in the N0 and N1 treatments.

**Table 1.** Soybean nodules numbers and nodules dry weight with different nitrogen application rates.

| Treatments | No.of Nodule Per Plant (no/Plant) | | | DM of Nodule Per Plant (g/Plant) | | |
|---|---|---|---|---|---|---|
| | SS | IS | AVE | SS | IS | AVE |
| N0 | 196 ± 6.02 b | 218 ± 4.35 a | 207 ± 17.91 A | 0.87 ± 0.15 a | 0.99 ± 0.16 a | 0.93 ± 0.15 A |
| N1 | 164 ± 9.45 b | 209 ± 3.51 a | 187 ± 13.47 B | 0.80 ± 0.09 a | 0.88 ± 0.02 a | 0.84 ± 0.07 A |
| N2 | 119 ± 11.2 b | 151 ± 14.4 a | 135 ± 21.16 C | 0.51 ± 0.01 a | 0.57 ± 0.11 a | 0.54 ± 0.08 B |
| ANOVE | | *p* | | | *p* | |
| N | | 0.00 | | | 0.00 | |
| C | | 0.00 | | | 0.11 | |
| N×C | | 0.03 | | | 0.93 | |

NOTE: N indicates nitrogen application rates, C indicates planting patterns; N×C shows the interaction between nitrogen application rates and planting patterns. Different lowercase letters indicate significant differences between single intercropping at the same nitrogen supply level, and different capital letters indicate significant differences between different nitrogen supply levels. The following table is the same. Data are presented as a Mean ± SD ($n = 5$).

### 3.3. Alpha Diversity and Composition of Nitrogen-Fixing Bacteria of Soybean Rhizosphere Soil

A total of 115396 reads were obtained from sequencing and 66.22% of them could be assigned. The Shannon index (Figure 2a) and Ace index (Figure 2b) of the nitrogen-fixing bacteria community in soybean rhizosphere soil were significantly influenced by the nitrogen application rates ($p < 0.01$). The Shannon index decreased when the nitrogen application rates increased from N0 to N1, while no significant difference was observed between N1 and N2 treatments. Intercropping maize significantly increased the Shannon index in the soybean rhizosphere compared to monocropping in N0 (0 kg N/ha) and N1 (40 kg N/ha) treatments. However, the Shannon index of intercropped soybean was significantly decreased by 1.05 times in N2 (80 kg N/ha) treatment when compared to the corresponding monocropping cultivation. Similar results were found for the ACE index in soybean rhizosphere soil. Nitrogen application significantly influenced the ACE index



($p < 0.05$), while planting patterns did not have a significant effect. However, a significant interaction was observed between nitrogen application and planting patterns on the ACE index ($p < 0.01$).

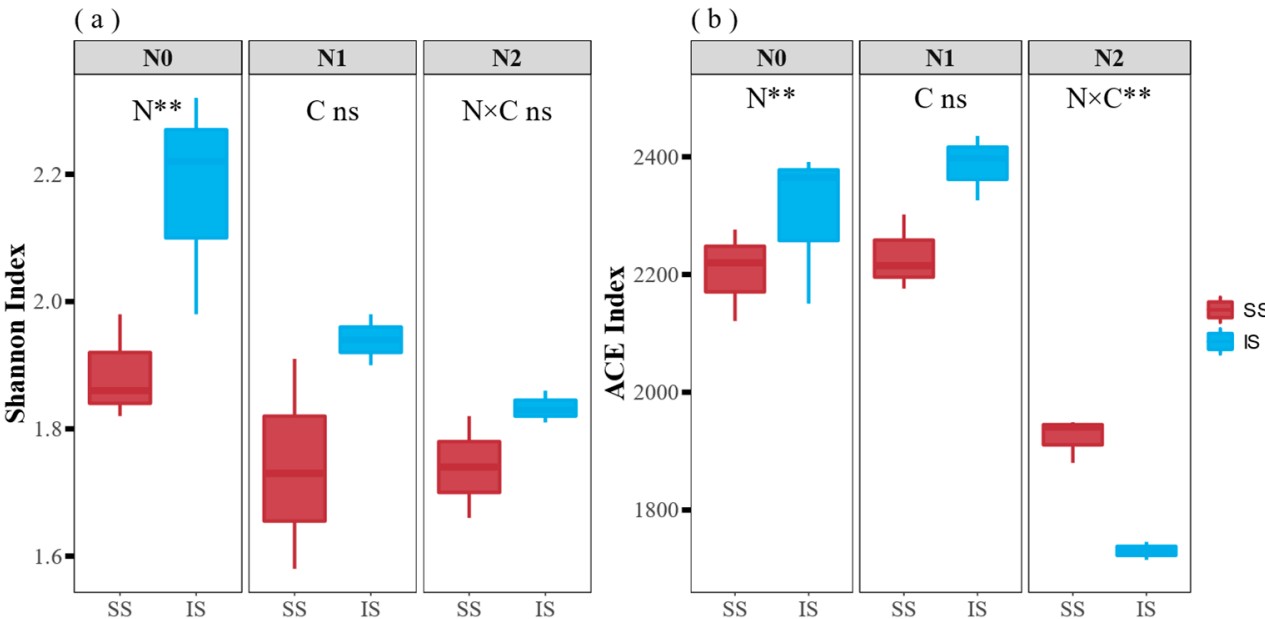

**Figure 2.** The Shannon index (**a**) and ACE index (**b**) of nitrogen-fixing bacteria were measured in soybean under different nitrogen levels. N indicates nitrogen application rates; C indicates planting patterns; N×C shows the interaction between nitrogen application rates and planting patterns by two ways ANOVA. ns and **, respectively, indicate that the difference is not significant, and difference is significant at levels of $p < 0.01$. Data are presented as a Mean ± SD ($n = 3$).

The top five dominant genera in soybean rhizosphere soil with more than 95% of total abundance were *Bradyrhizobium*, *Skermanella*, *g_unclassified_p_Proteobacteria*, *g_unclassified_k __norank_d__Bacteria*, and Azohydromonas (Figure 3a). Among these, *Bradyrhizobium* was the most abundant genus accounting for more than 70% of the total abundance. The lowest *g_unclassified_k__norank_d__Bacteria* abundance is only about 3% of the total abundance. The abundance of Azohydromonas was significantly different under different nitrogen levels when single cropping was not considered. It was found that the abundance of Azohydromonas under the N0 treatment was significantly increased by 2.62 times compared with the N2 treatment.

Different nitrogen application rates and intercropping did not significantly affect *Bradyrhizobium*, *Skermanella*, *g_unclassified_p_Proteobacteria,* and *g_unclassified_k__norank_d__ Bacteria* in soybean rhizosphere, but *Azohydromonas* was significantly affected by nitrogen application rates ($p < 0.01$). *g_unclassified_k__norank_d__Bacteria* was significantly affected by the planting pattern ($p < 0.05$) and the interaction between nitrogen application and planting pattern ($p < 0.01$). The highest abundance of *Azohydromonas* was found in the N0 treatment (0 kg N/ha) and the lowest in N2 (80 kg N/ha). The rhizosphere *Azohydromonas* abundance of intercropping soybean was significantly increased by 35%, 7%, and 42% compared with that of monoculture under N0, N1, and N2 nitrogen levels. When nitrogen application rates were 0 kg/ha or 40 kg/ha, *g_unclassified_k__norank_d__Bacteria* abundance was higher compared to monocropping. However, in the 80 kg N/ha application rate, it was lower than intercropping.

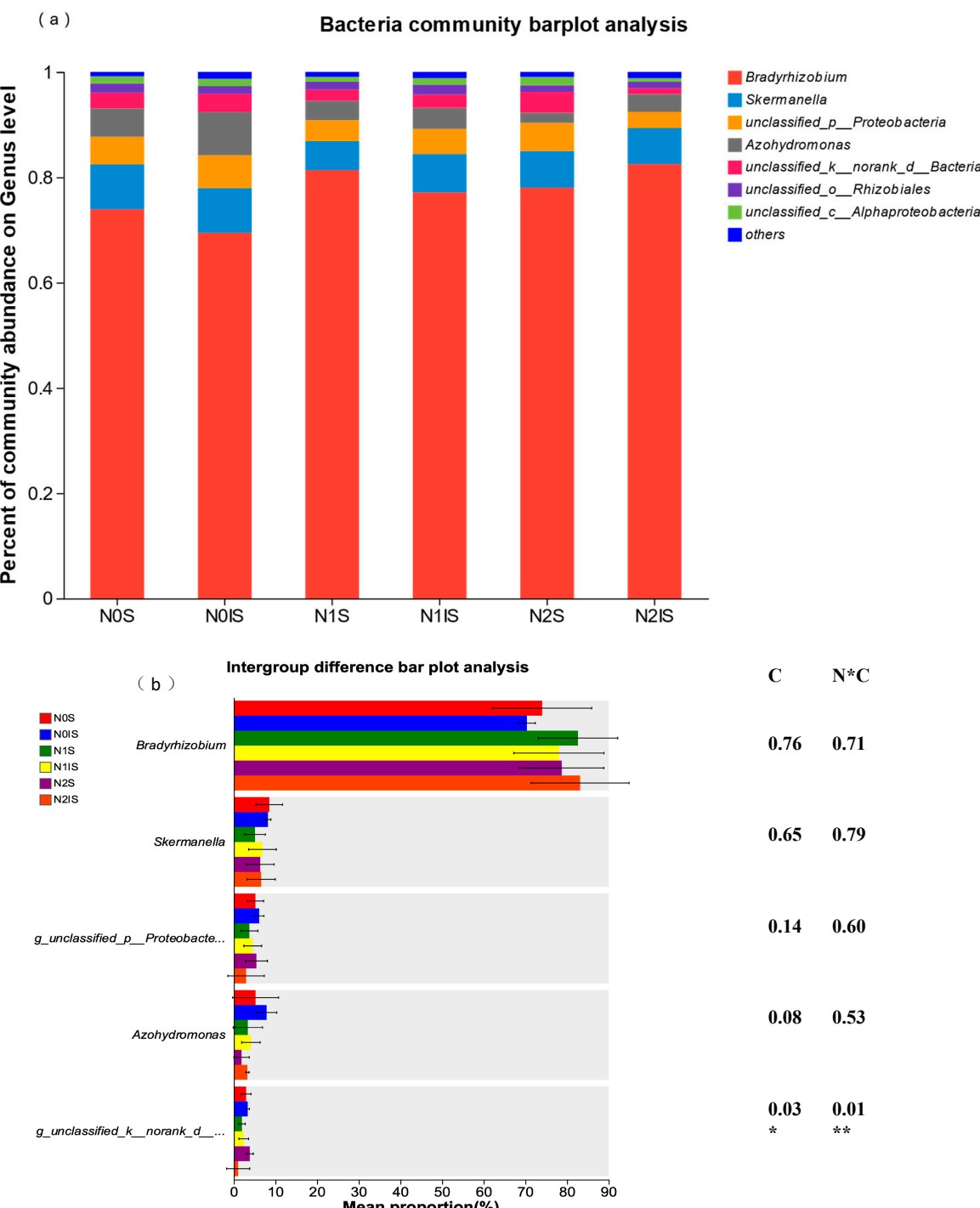

**Figure 3.** The soybean nitrogen-fixing bacteria community composition (**a**), differences between groups (**b**) under different nitrogen application levels. N indicates nitrogen application rates; C indicates planting patterns; N×C shows the interaction between nitrogen application rates and planting patterns by two ways ANOVA. ns and * and **, respectively, indicate that the difference is not significant, and difference is significant at levels of $p < 0.05$ and $p < 0.01$. Data are presented as a Mean ± SD ($n = 3$).

### 3.4. Relationship between the Nitrogen-Fixing Bacteria Community and Soil Physicochemical Characteristics

The soybean rhizosphere soil physicochemical characteristics were affected by nitrogen application rates and planting patterns (Table 2). Specifically, nitrogen content (AN), available phosphorus (AP), and pH were significantly influenced by nitrogen application rates ($p < 0.01$), with an increase observed as nitrogen application rates increased. The available phosphorus (AP) was also significantly affected by the cultivation pattern ($p < 0.05$), with the soil AP in the monocropping system being significantly higher compared to intercropping. However, no significant difference was found in available potassium (AK) and organic matter (OM) among different nitrogen treatments and cultivation patterns.

**Table 2.** Soybean soil physicochemical characteristics of different treatments with different nitrogen application rates.

| Treatments | AN (mg/kg) | | | AP (mg/kg) | | | AK (mg/kg) | | | pH | | | OM (g/kg) | | |
|---|---|---|---|---|---|---|---|---|---|---|---|---|---|---|---|
| | SS | IS | AVE | SS | IS | AVE | SS | IS | AVE | SS | IS | AVE | SS | IS | AVE |
| N0 | 132 ± 1.52 a | 131 ± 4.35 a | 131 ± 3.01 B | 71 ± 0.55 b | 70 ± 0.26 a | 70 ± 0.48 B | 141 ± 1.07 a | 143 ± 2.50 a | 142 ± 1.85 A | 6.66 ± 0.08 a | 6.60 ± 0.03 a | 6.63 ± 0.06 B | 30 ± 0.65 a | 30 ± 1.03 a | 30 ± 0.77 A |
| N1 | 132 ± 3.51 a | 132 ± 1.01 a | 132 ± 2.31 B | 71 ± 0.14 b | 70 ± 0.39 a | 71 ± 0.32 A | 141 ± 1.55 a | 143 ± 1.12 a | 142 ± 1.39 A | 6.67 ± 0.08 a | 6.65 ± 0.08 a | 6.66 ± 0.07 AB | 31 ± 0.65 a | 31 ± 1.79 a | 31 ± 0.68 A |
| N2 | 155 ± 2.02 a | 154 ± 4.93 a | 155 ± 3.37 A | 72 ± 0.56 b | 71 ± 0.05 a | 71 ± 0.57 A | 142 ± 0.57 a | 142 ± 2.95 a | 142 ± 1.90 A | 6.75 ± 0.02 a | 6.72 ± 0.02 a | 6.73 ± 0.02 A | 31 ± 0.28 a | 31 ± 0.97 a | 31 ± 0.70 A |
| ANOVE | | *p* | | | *p* | | | *p* | | | *p* | | | *p* | |
| N | | 0.00 | | | 0.01 | | | 0.95 | | | 0.03 | | | 0.29 | |
| C | | 0.67 | | | 0.03 | | | 0.40 | | | 0.25 | | | 0.35 | |
| N×C | | 0.95 | | | 0.59 | | | 0.71 | | | 0.84 | | | 0.89 | |

NOTE: N indicates nitrogen application rates, C indicates planting patterns; N×C shows the interaction between nitrogen application rates and planting patterns. Different lowercase letters indicate significant differences between single intercropping at the same nitrogen supply level, and different capital letters indicate significant differences between different nitrogen supply levels. The following table is the same. Data are presented as a Mean ± SD ($n = 3$).

RDA analysis by "vegan package "was conducted to investigate the relationship between soil environmental factors and nitrogen-fixing bacterial communities in the soybean rhizosphere. The first two RDA axes explained 16.86% of the variance in bacterial community composition (Figure 4a). In the RDA analysis, the eigenvalues of DCA1 and DCA2 were 0.0473 and 0.0131, respectively. Based on the RDA, AP, AN, and pH were found to have a higher contribution rate to RDA1, while OM and AK had a higher contribution to RDA2. However, pH, AN, AP, OM, and AK did not significantly affect the nitrogen-fixing bacterial communities (Table 3). To further examine the correlation between soil physicochemical characteristics and nitrogen-fixing bacterial communities, we conducted the Mantel test using the "ggcor" package. The results indicated no significant relationship between pH, AN, AP, OM, AK, and the nitrogen-fixing bacterial communities in the soybean rhizosphere soil (Figure 4b).

**Table 3.** RDA analysis results of nitrogen-fixing bacteria communities in soybean rhizosphere soil with different nitrogen application rates.

| Soil Characteristic | RDA1 | RDA2 | $r^2$ | *p* |
|---|---|---|---|---|
| AN | 0.8445 | 0.5356 | 0.2044 | 0.176 |
| OM | 0.3247 | 0.9458 | 0.0527 | 0.683 |
| AP | 0.8044 | 0.594 | 0.2444 | 0.114 |
| AK | −0.5725 | −0.8199 | 0.0502 | 0.681 |
| pH | 0.9566 | 0.2915 | 0.0672 | 0.577 |

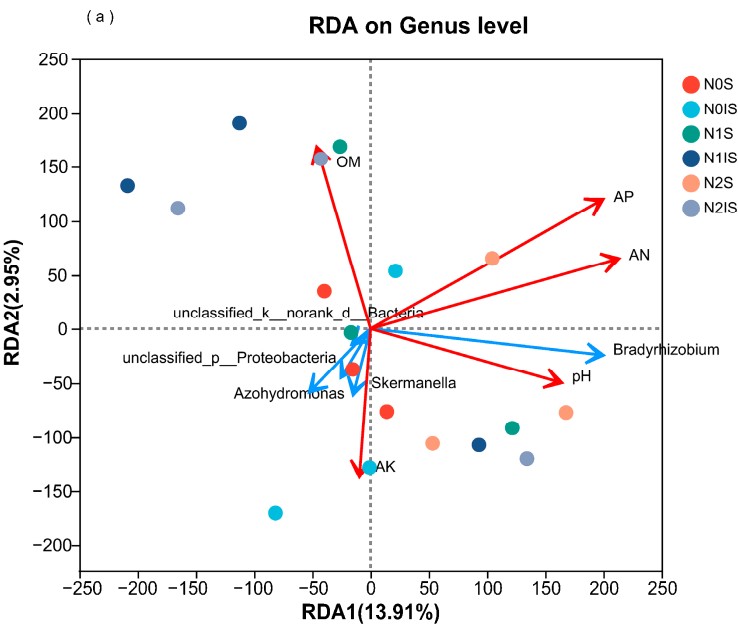

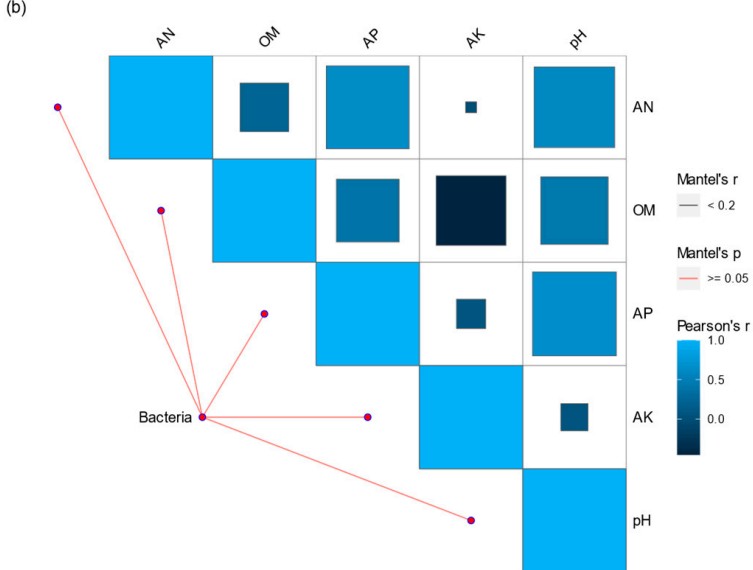

**Figure 4.** RDA (**a**) and Mantel test (**b**) analysis of soil characteristics of soybean under different nitrogen application levels. Color gradient and size represent Pearson correlation coefficients of physicochemical factors. The edge width represents the Mantel R statistic of the corresponding correlation coefficient, and the color indicates the Mantel *p* statistic. AN, soil available nitrogen; OM, soil organic matter; TN, soil total nitrogen content; AP; Soil available phosphorus content; AK, soil available potassium.

### 3.5. nifH Gene Copies of Nitrogen-Fixing Bacteria in Soybean and Maize Rhizosphere Soil

The *nifH* gene abundance was significantly affected by nitrogen application ($p < 0.01$) but not significantly affected by planting patterns ($p > 0.05$) (Figure 5a). The highest *nifH* gene copy number ($6.25 \times 10^7$ copies/g soil) was observed in the N1 treatment and the lowest in the N2 treatment ($3.54 \times 10^7$ copies/g soil). Intercropping with maize increased the *nifH* gene copies in the N0 and N1 treatments by 11% and 13%, respectively, but it resulted in a decrease of 23% in the N2 treatment compared to monocropping.

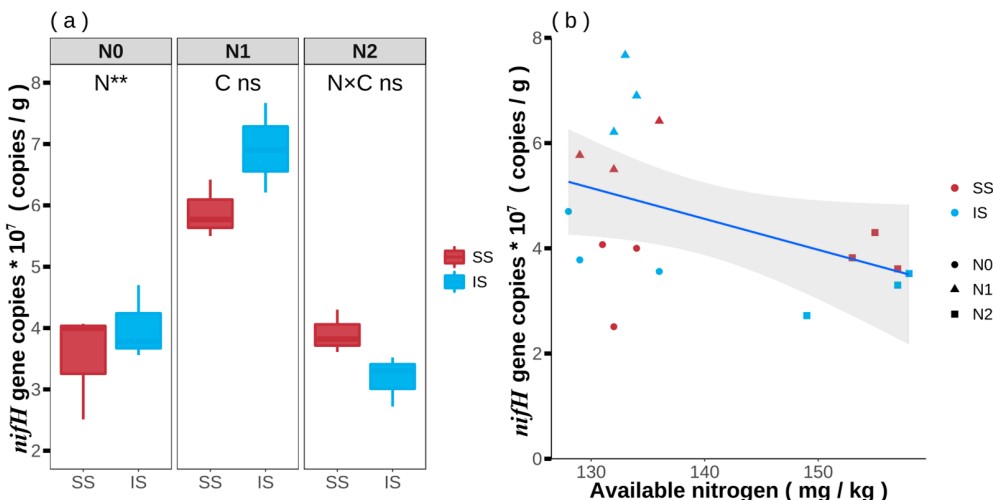

**Figure 5.** The *nifH* gene copies in soybean rhizosphere soil with different nitrogen levels ($\times 10^7$) (**a**) and their correlation with soil available nitrogen (**b**) with different nitrogen application rates. N indicates nitrogen application rates; C indicates planting patterns; N×C shows the interaction between nitrogen application rates and planting patterns by two ways ANOVA. ns and **, respectively, indicate that the difference is not significant, and difference is significant at levels of $p < 0.01$. Data are presented as a Mean $\pm$ SD ($n$ = 3).

### 3.6. Correlation nifH Gene Copies and Soil Available and Biological Quantification of Nitrogen Fixation

To further elucidate the relationship between soil available nitrogen, *nifH* gene copies, and nitrogen fixation, we conducted Pearson correlation analyses. The results showed a significant negative correlation ($p < 0.05$) between soil available nitrogen and *nifH* gene copies in soybean rhizosphere soil, with a linear relationship expressed with the equation y = $-0.0606x + 13.02$ and an $r^2 = 0.2243$ (Figure 5b).

## 4. Discussion

### 4.1. Nitrogen Fixation of Intercropped Soybean Affected by Nitrogen Application Rates

Several studies have shown that intercropping systems can improve the nitrogen fixation efficiency of legumes, consistent with the results of our research (Figure 1a) [36]. When soybean and maize are intercropped, the roots of maize can grow into the area where soybean roots grow, leading to competition for available soil nitrogen [8]. As a result, soybean roots are in a low nitrogen environment, which can stimulate nitrogen fixation from the air and thus improve the soybean nitrogen fixation efficiency. In addition, proper nitrogen application could also improve legume growth [22]. We obtained similar results, showing that soybean nitrogen fixation was the highest in the N1 treatment (40 kg N/ha) (Figure 1b). The increase in nitrogen fertilizer in intercropped soybean significantly negatively affected the nodule number and dry weight (Table 1). Nodules not only serve as sites for atmospheric nitrogen (N2) fixation but also provide an energy source for rhizobia [37]. This decrease in nodule formation ultimately impacted soybeans' ability to fix nitrogen. Our results suggest that maize and soybean intercropping did not completely enhance soybean nitrogen fixation. In fact, the nitrogen fixation rates in the N0 and N2 intercropping treatments were still 1.08 and 1.19 times lower, respectively, compared to monocropping. However, in the N1 intercropped soybean treatment, nitrogen fixation was 1.04 times higher than monocropping. The lower nitrogen fixation in the intercropping system compared to monoculture was attributed to the competition from intercropped maize, which resulted in a decrease in soybean biomass in the intercropping system [38]. Our study also yielded similar results, as the soybean biomass under monocropping was significantly higher than that under intercropping.

The nitrogen fixation efficiency is closely related to the soybean nodule number, size, and dry weight. In our study, nodules were crucial in improving nitrogen fixation efficiency. After intercropping soybean with maize and applying the appropriate nitrogen fertilizers, the soybean's nitrogen fixation was enhanced. Our results indicated that nitrogen fixation efficiency in the N0 treatment was the highest, but nitrogen fixation in the N1 treatment was found to be the highest among all the treatments. Therefore, we suggest using 40 kg N/ha nitrogen fertilizer to maximize the soybean–maize intercropping system productivity.

### 4.2. Diversity of Nitrogen-Fixing Bacteria in the Rhizosphere Soil of Intercropped Soybean Affected by N Application

Nitrogen fixation and utilization in soil ecological environments are closely related to the nitrogen-fixing bacterial community structure and diversity. Nitrogen-fixing bacteria play an essential role in the nitrogen cycle by converting atmospheric nitrogen into a form available to plants [39]. The microbial diversity index is a crucial indicator of soil microbial diversity and can also reflect the quality of farmland ecosystems [40]. Intercropping practices can alter the diversity of nitrogen-fixing microorganisms [41]. In our study, we observed that intercropping of soybean resulted in a significantly higher Shannon index than monocropping (Figure 2a,b).

Previous studies have shown that intercropping can have varying effects on the nitrogen-fixing bacteria communities' diversity depending on the specific crop combinations and soil conditions. For example, cassava/peanut intercropping increased microbial diversity compared to peanut monoculture [42], while legume/oat intercropping increased the diversity of oat nitrogen-fixing bacteria communities [43]. However, intercropping may not significantly affect the Shannon index in some cases, such as in a maize/soybean/cotton intercropping system [44]. These discrepancies may be due to differences in soil environments and the effects of varying nitrogen application levels [45]. Therefore, it is important to consider the specific crop combinations and soil conditions when evaluating the impact of intercropping on nitrogen-fixing bacteria diversity.

In addition, different nitrogen application rates significantly impacted the nitrogen-fixing bacteria diversity in soybean. An appropriate nitrogen fertilization regime can increase soil bacterial diversity and richness [46]. However, previous studies have shown that a high nitrogen application level (100 kg/ha) reduced bacterial community diversity and richness in soybean rhizosphere [47], consistent with the results of our study. Therefore, high nitrogen application can result in a decrease in the rhizosphere microbial diversity in legumes.

Based on the species taxonomic analysis, the nitrogen-fixing bacteria community composition in soybean rhizosphere soil under different nitrogen application levels was investigated. A total of seven bacterial genera were detected in the soybean rhizosphere soil, with *Bradyrhizobium* being the dominant genus (Figure 3a). The nitrogen-fixing bacteria abundance in soybean was affected by both intercropping and nitrogen application.

The abundance of *g_unclassified_k__norank_d__Bacteria* in the soybean rhizosphere was significantly influenced by the planting patterns ($p < 0.05$), as shown in Figure 3b. Intercropping has been shown to change the composition of nitrogen-fixing microorganisms and increase the nitrogen-fixing bacteria species diversity [18], in agreement with our results. As the dominant genus, *g_unclassified_k__norank_d__Bacteria* was significantly more abundant in intercropping compared to monoculture under the N0 and N1 treatments. This suggests that intercropping can improve the nitrogen-fixing bacteria community composition in the appropriate nitrogen application.

Furthermore, the nitrogen application rate significantly impacted the community structure of nitrogen-fixing bacteria in the soybean/maize intercropping system. For instance, the abundance of Azohydromonas in the soybean rhizosphere decreased with the increase in nitrogen application rate. This finding is consistent with previous studies indicating that mycorrhizal fungi have a weaker ability to infect plant roots in high-nitrogen environments, which can ultimately lead to a reduction in soil microbial activity [48,49].

Thus, it is crucial to apply the appropriate nitrogen fertilization rates to maintain the stability and diversity of the soil microbial community.

Our study showed that nitrogen application rates and planting patterns significantly affected soil physicochemical characteristics, with soil AN, AP, and pH significantly affected by nitrogen application rates. Similar research has shown that farmland management can significantly affect the soil's physicochemical properties and biodiversity [50], with AN being the crucial factor affecting microbial diversity [51] and pH significantly affecting nitrogen-fixing microorganisms [52]. Changes in soil physicochemical properties due to different nitrogen application rates and planting patterns can alter the nitrogen-fixing bacteria community structure. However, in our study, the RDA results indicated that the soil pH and AN content had no significant effects on the soybean rhizosphere soil.

### 4.3. NifH Gene Copies in the Rhizosphere Soil of Intercropped Soybean

In different soil ecological environments, the *nifH* gene diversity was significantly different [53]. Planting methods have also been shown to affect the *nifH* gene copy number, as demonstrated by [52], who studied forest soil under different cultivation modes. Moreover, the soybean–maize intercropping increased *nifH* gene copies by 11% and 13% in N0 and N1 treatments compared with monoculture, respectively (Figure 4a). This suggests that intercropping can increase *nifH* gene copies in appropriate nitrogen application. It has been reported that the *nifH* gene expression, responsible for nitrogen fixation, is affected by soil nitrogen concentration [54,55]. Correlation analysis indicated that the *nifH* gene copy number was significantly negatively correlated with soil-available nitrogen (Figure 5b). The highest *nifH* gene copies were observed in the N1 intercropping treatment of soybean. This is likely because maize requires a significant amount of nitrogen during the tasseling period, intensifying competition for soil nitrogen. It stimulated soybean to regulate root exudates, altering the nitrogen-fixing bacteria community to enhance its nitrogen-fixing function to meet its own needs. As shown from previous studies, reducing nitrogen levels can improve the nitrogen fixation intensity and nitrogen-fixing bacteria number in maize and soybean intercropping systems [18].

Excessive nitrogen can have a negative impact on the activity of nitrogen-fixing microorganisms [56]. Our study found that the *nifH* gene copy numbers, an indicator of nitrogen fixation, were significantly affected by different nitrogen application rates in soybean rhizosphere soil. The highest *nifH* gene copy numbers were observed in the N1 treatment in soybean, indicating that appropriate nitrogen application can increase gene copies. However, excessive nitrogen application can inhibit the activity of nitrogen-fixing bacteria and decrease the abundance of the nifH gene, which is consistent with previous research results [54,57]. Moreover, excessive nitrogen fertilizer can lead to soil acidification, which is not conducive to the growth of microorganisms [58].

### 5. Conclusions

Nitrogen-fixing bacteria diversity of intercropped soybean rhizosphere was significantly affected by nitrogen application rates and planting patterns. *Bradyrhizobium* was the dominant genus. The nitrogen-fixing efficiency, quantity of *nifH* gene copies, and nodules number of intercropped soybean were increased under different nitrogen application rates due to the decrease of AN in the rhizosphere of soybean. The Pearson correlation analysis indicated that the available nitrogen content in the soil significantly negatively impacted the abundance of *nifH* genes in the soybean rhizosphere soil. These results provide an important idea for improving the nitrogen-fixing efficiency and reducing the use of chemical nitrogen fertilizer in legume/Gramineae intercropping systems.

**Author Contributions:** Z.C.: Experiment design, Writing—original draft preparation, sampling and analysis. L.M.: investigation, funding acquisition. T.Y.: Writing—reviewing. Y.L.: Writing—original draft. Y.Z.: Data analysis. S.L.: Editing, project administration, supervision, funding acquisition. All authors have read and agreed to the published version of the manuscript.

**Funding:** This research was funded by National Natural Science Foundation of China (32171547).

**Data Availability Statement:** Most of the collected data are contained in the tables and figures in the manuscript.

**Conflicts of Interest:** The authors declare no conflict of interest.

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
