# Peer review of "Changes in Soil Rhizobia Diversity and Their Effects on the Symbiotic Efficiency of Soybean Intercropped with Maize"

_agronomy, doi:10.3390/agronomy13040997_

Round 1

Reviewer 1 Report

Manuscript ID: agronomy-2281747

Title: Change of Soil Rhizobia Diversity and its effects on the Symbiotic Efficiency of Soybean Intercropped with Maize

Authors: Zeyu Cheng , Lingbo Meng , Tengjiao Yin , Ying Li , Yuhang Zhang , Shumin Li *

General and Major Comments

The authors studied effects of nitrogen fertilization and intercropping with maize on nitrogen-fixing efficiency, nitrogen-fixing bacterial diversity, nifH gene copy number, and soil physicochemical characteristics. Then they showed that higher nitrogen application tended to decrease the nitrogen-fixing efficiency, the number of nodules, and the nitrogen-fixing bacterial diversity. The authors found that the copy number of nifH in the condition of medium nitrogen application (40kg N ha-1) was higher than that in the condition of no and high nitrogen application.

The manuscript contained interesting results and showed information about the effects of nitrogen fertilization and intercropping on nitrogen fixing bacteria. It is particularly interesting to note that the impact of the two factors was investigated in combination. However, the manuscript needs to be revised in its entirety, including the following, before publication.

1. I would like to suggest to the authors rewrite manuscript because there were several grammatical and syntactical errors and notational errors. Revision by native English speaker should be desirable.

2. A detailed description of the methods and results (including figures) for the genetic analysis (L161-174, 244-312) is needed. In particular, the nitrogen-fixing bacterial community (Figure 3).

3. Some figures and text did not match (For example, Figure 2, Line 248-, and 362-; Figure 3a and Line 269-; Figure 4a and Line 343).

4. The authors concluded that medium nitrogen application was appropriate based on the result of the copy number of nifH. However, from the perspective of nitrogen-fixing efficiency, no fertilizer application could be considered appropriate. In addition, because of the lack of yield data, which is the most important, it is difficult to conclude the appropriate amount of nitrogen application. I think the authors need to clarify “appropriate” and “proper” in the manuscript. (If the authors measured yields of soybean and maize, they should put them up. Discussion of the relevance of the yields and other parameters would greatly improve this paper.)

5. I think that the authors should not directly associate nifH copy number with nitrogen-fixing activity in the manuscript because they did not examine the expression of nifH and they measured the number of nifH in the soil. The authors examined nitrogen-fixing activity by measure of 15N in soybean plants, therefore nitrogen-fixing activity in this study contained fixed nitrogen in nodule.

Some Minor Comments

1. Statements that imply that soybean fixes nitrogen should be avoided (Line 71, 208, and 215 etc...).

2. It is not appropriate to describe unclassified_XXXXXXXX as one genus (Line 383 and 391).

3. Consents in Table S1, 2, and 3 should be summarized and presented in the main manuscript because the authors discussed soil physicochemical characteristics in details.

4. I could not understand why there were 18 points in Figure 4b and 4c because I understand that 5 soybean plants were taken (Line 133) and there were three nitrogen fertilization application rates (123-). Similarly, I did not understand what “9 treatments” indicated (Line 128). I understood that the experiment was conducted under three nitrogen fertilization application rates (N0, N1, and N2) and two planting patterns (soybean monocropping and maize/soybean intercropping). Thus, I thought it was 6 treatments and the figures and tables also appear to be 6 conditions.

5. There was not the result of nitrogen fixing bacterial community of maize rhizosphere soil in the manuscript although the authors mentioned it (Line 407).

I hope these comments will be helpful.

Author Response

Responses to the comments

Manuscript Number: agronomy-2281747

We thank you and the reviewers for the very helpful comments on our manuscript ‘ Changes in Soil Rhizobia Diversity and their effects on the Symbiotic Efficiency of Soybean Intercropped with Maize(agronomy-2281747)’. We have carefully considered all comments and thoroughly revised the text. The English of the whole text has been modified thoroughly by a native speaker. We also have prepared all of the items required by the submission system. We hope the revised manuscript is suitable for publication in Agronomy.

All our responses to reviewers are detailed in blue. Changes made in the manuscript are highlighted in red. The revision has been developed in consultation with all coauthors, and each author has given approval to the final form of this revision.

Prof. Shumin Li

Resource and Environmental College, Northeast Agricultural University

1.I would like to suggest to the authors rewrite manuscript because there were several grammatical and syntactical errors and notational errors. Revision by native English speaker should be desirable.

Response: We regret there were problems with the English. The paper has been carefully revised by a professional language editing service to improve the grammar and readability.

2. A detailed description of the methods and results (including figures) for the genetic analysis (L161-174, 244-312) is needed. In particular, the nitrogen-fixing bacterial community (Figure 3).

Response: Thanks for your suggestion. I've added a more detailed genetic analysis methods(L166-199),We also added the results of detailed genetic analysis especially the results of the community of nitrogen fixing bacteria.(Line289-368)

3. Some figures and text did not match (For example, Figure 2, Line 248-, and 362-; Figure 3a and Line 269-; Figure 4a and Line 343).

Response: I apologize for this mistake. Figures 2, Figure 3a, and Figure 4a have been modified. The Figure already matches the text in the manuscript.(Figure2 and Line290;Figure3a and Line305;Figure5a and Line357)

4. The authors concluded that medium nitrogen application was appropriate based on the result of the copy number of nifH. However, from the perspective of nitrogen-fixing efficiency, no fertilizer application could be considered appropriate. In addition, because of the lack of yield data, which is the most important, it is difficult to conclude the appropriate amount of nitrogen application. I think the authors need to clarify “appropriate” and “proper” in the manuscript. (If the authors measured yields of soybean and maize, they should put them up. Discussion of the relevance of the yields and other parameters would greatly improve this paper.)

Response: Thanks for your suggestion. Indeed, in this manuscript, nifH gene copy number in N1 treatment is the highest. N fixation  efficiency in N0 treatment is the highest. However, soybean biomass at N0 was lower than that of N1 treatment, so N fixation in N1 is higher than the other treatment. we added soybean biomass data in Figure1d.(Line258-264,395-398)

5. I think that the authors should not directly associate nifH copy number with nitrogen-fixing activity in the manuscript because they did not examine the expression of nifH and they measured the number of nifH in the soil. The authors examined nitrogen-fixing activity by measure of 15N in soybean plants, therefore nitrogen-fixing activity in this study contained fixed nitrogen in nodule.

Response: Thanks for your suggestion. According to the opinion of another reviewer, we deleted this part of content and (Figure 4c)

Some Minor Comments1. Statements that imply that soybean fixes nitrogen should be avoided (Line 71, 208, and 215 etc...).

Response: Thanks for your suggestion. We avoid using statements that imply that soybean fixes nitrogen(Line79,233,240 etc...)

2. It is not appropriate to describe unclassified_XXXXXXXX as one genus (Line 383 and 391).

Response: Thanks for your suggestion. We amend the unclassified_k__norank_d__Bacteria and unclassified_p_Proteobacteria to g_unclassified_p_Proteobacteria and g_unclassified_k__norank_d__Bacteria. To indicate that they belong to some unclassified genus of bacteria(Line432,436)

3. Consents in Table S1, 2, and 3 should be summarized and presented in the main manuscript because the authors discussed soil physicochemical characteristics in details.

Response: Thanks for your suggestion. We move Table S1 into the main manuscript as Table 2; We move Table S2 to Table 3 in the main manuscript and described the results in detail(Line340-343); We move Table S3 into the main manuscript as Figure 4b.

4. I could not understand why there were 18 points in Figure 4b and 4c because I understand that 5 soybean plants were taken (Line 133) and there were three nitrogen fertilization application rates (123-). Similarly, I did not understand what “9 treatments” indicated (Line 128). I understood that the experiment was conducted under three nitrogen fertilization application rates (N0, N1, and N2) and two planting patterns (soybean monocropping and maize/soybean intercropping). Thus, I thought it was 6 treatments and the figures and tables also appear to be 6 conditions.

Response: I apologies for this mistake. There are two planting patterns in this experiment, three nitrogen levels and six treatments with three replicates in each treatment. There are altogether 18 plots. I have corrected them in the materials and methods(Line131-132). Five soybean plants were taken  to measure 15 N abundance and nitrogen concentration analysis(Line 137-141)

5. There was not the result of nitrogen fixing bacterial community of maize rhizosphere soil in the manuscript although the authors mentioned it (Line 407).

Response: I apologies for this mistake. This manuscript focuses on changes in rhizosphere nitrogen fixation bacteria diversity between monoculture and intercropping soybeans, so we deleted this part.(Line 448-456)

The revised manuscript is attached

Reviewer 2 Report

The research entitled (Change of Soil Rhizobia Diversity and its effects on the Symbiotic Efficiency of Soybean Intercropped with Maize) investigates the nitrogen fixation bacterial communities, N fixing efficiency, and their relationships under three nitrogen fertilization application rates.

1.     Author should clearly mention the novelty of this study by point wise.

2.     why did the authors choose these specific nitrogen concentrations (N0, N1 117 and N2 were 0 kg N ha-1, 40kg N ha-1 and 80kg N ha-1)?

3.     Authors need to include the environmental conditions during the experiment period.

4.     Figure 4C, access title needs correction.

5.     The quality of the figures should be improved.

6.     In lines 259-260; In the rhizosphere of soybean, the top five dominant genera were Bradyrhizobium, Skermanella, unclassified_p_Proteobacteria, unclassified_k__norank_d__Bacteria, and Azohydromonas (Figure 3a).  What is the meaning of _p_, _k__,_d__?

7.     Discussion part needs to fortify with more recent references.

8.     The reference list needs to check for the references page numbers and style.

Author Response

Responses to the comments

Manuscript Number: agronomy-2281747

We thank you and the reviewers for the very helpful comments on our manuscript ‘ Changes in Soil Rhizobia Diversity and their effects on the Symbiotic Efficiency of Soybean Intercropped with Maize(agronomy-2281747)’. We have carefully considered all comments and thoroughly revised the text. The English of the whole text has been modified thoroughly by a native speaker. We also have prepared all of the items required by the submission system. We hope the revised manuscript is suitable for publication in Agronomy.

All our responses to reviewers are detailed in blue. Changes made in the manuscript are highlighted in red. The revision has been developed in consultation with all coauthors, and each author has given approval to the final form of this revision.

Prof. Shumin Li

Resource and Environmental College, Northeast Agricultural University

1.Author should clearly mention the novelty of this study by point wise.

Response: Thank you for this valuable feedback. The novelty of our study was that we investigated the change of nitrogen-fixing bacteria and nifH genes of intercropped soybean compared with monoculture, and isotope 15N method was used to measure nitrogen fixation efficiency of intercropped soybean at different nitrogen supply levels. We clarified the novelty in introduction. See (Line97-100)

2. Why did the authors choose these specific nitrogen concentrations (N0, N1 117 and N2 were 0 kg N ha-1, 40kg N ha-1 and 80kg N ha-1)?

Response: Thanks for your suggestion. The nitrogen level of this experiment was set according to the conventional nitrogen application level of soybean in the black soil area of Northeast China. 40kg/ha was conventional nitrogen application rate, 80kg/ha was excessive nitrogen application rate, and 0kg/ha was control treatment.

3.     Authors need to include the environmental conditions during the experiment period.

Response: Thanks for your suggestion. I added the environmental conditionsof the experiment period in the materials and methods (Lines 110-113).

4.     Figure 4C, access title needs correction.

Response: Thanks for your suggestion. According to the comments of other reviewers, we deleted Figure4c.

5.     The quality of the figures should be improved.

Response: Thanks for your suggestion. Quality of The Figure1,Figure2,Figure 3 and the Figure 4 has been improved.

6. In lines 259-260; In the rhizosphere of soybean, the top five dominant genera were Bradyrhizobium, Skermanella, unclassified_p_Proteobacteria, unclassified_k__norank_d__Bacteria, and Azohydromonas (Figure 3a).  What is the meaning of _p_, _k__,_d__?

Response: Thanks for your question. _p is phylum, _k is kingdom, _d is domain. Unclassified_k__norank_d__Bacteria means that the genus has not been classified in Bacteria domain. unclassified_p_Proteobacteria means that the genus has not been classified in Proteobacteria. We amend the unclassified_k__norank_d__Bacteria and unclassified_p_Proteobacteria to g_unclassified_p_Proteobacteria and g_unclassified_k__norank_d__Bacteria. (Line303-305,432,436)

7.     Discussion part needs to fortify with more recent references.

Response: Thanks for your suggestion. References in the discussion part has been updated with more recent references. See (Line 372,406,417,465,475)

Reference

Chamkhi, I., S. Cheto, J. Geistlinger, Y. Zeroual, L. Kouisni, A. Bargaz and C. Ghoulam (2022). "Legume-based intercropping systems promote beneficial rhizobacterial community and crop yield under stressing conditions." Industrial Crops and Products 183.

Din, I., H. Khan, N. Ahmad Khan and A. Khil (2021). "Inoculation of nitrogen fixing bacteria in conjugation with integrated nitrogen sources induced changes in phenology, growth, nitrogen assimilation and productivity of wheat crop." Journal of the Saudi Society of Agricultural Sciences 20(7): 459-466.

Li, Y., F. Pan and H. Yao (2019). "Response of symbiotic and asymbiotic nitrogen-fixing microorganisms to nitrogen fertilizer application." Journal of Soils and Sediments 19(4): 1948-1958.

Lankau, R. A., I. George and M. Miao (2022). "Crop performance is predicted by soil microbial diversity across phylogenetic scales." Ecosphere 13(5).

Koocheki, A., H. Solouki and S. Karbor (2016). "Study of ecological aspects of Sesame (Sesamum indicum L.) and Mung Bean (Vigna radiata L.) intercropping in weed control." Iranian Journal of Pulses Research 7: 27-44.

8.     The reference list needs to check for the references page numbers and style.

Response: Thanks for your suggestion. I have checked the style and number of pages of the completed references.

The revised manuscript is attached

Reviewer 3 Report

The work submitted by the authors is relevant to the study of microbiota in intercrops, particularly rhizobia, which naturally associate with legumes. Before acceptance, I consider important changes should be made to the manuscript.

Suggestions are detailed below:

Please check the English with a native reviewer, as many sentences are very long and repetitive.

Abstract

Please specify nitrogen concentrations after mentioning the treatment.

 L20-21, it is not understood to which variables the percentages correspond. 

L22 Specify with respect to which treatment an increase of nitrogen-fixing bacteria diversity was observed

L25 Specify what AN and other abbreviations mean.

L26 Conclusions should be limited to the effects of intercropping and nitrogen supply on rhizobia diversity.

Introduction 

Add data from the bean/maize system, where there is already work on microbial diversity and its transcriptomic responses.

L73, 76 Review the correct way to cite in-text references 

L85, what do you mean by gene replication? Rephrase the idea please

L86, be sure to spell scientific names correctly.

I recommend first explaining aspects of bacterial nitrogenase regulation and then the findings associated with N-fertilization.

Include more references to work on the effect of rhizobial diversity and nitrogen fixation in soybeans in response to high nitrogen levels, either in the introduction or the discussion.

Materials and methods

In the experimental design appeared results of soil properties, those tests were performed by you or are previously reported?  Specify and do not include results in the methodology.

Mention the methods used for each chemical determination and its reference.

L150, is the dry weight of the plant or nodules? Be more specific about which sample was analyzed.

162, specify where the DNA was extracted from. Soil, rhizosphere or from bacteria?

L163 Include more information about the DNA extraction, include the brand of the kit and if the manufacturer's recommendations were followed. I could not find it on the web.

L165, why were these primers selected, what region do they amplify and for which group of diazotrophs is it effective. What is the size of the PCR product? Were adapters included for the sequencing platform, please include the sequence in supplementary material?

L173 Could you please add how you performed the copy number calibration and standard curve? Include how many replicates were done for qPCR and normalization methods.

L176, explain in an orderly fashion how 15N was determined, from sample collection to concentration calculations.

L196 Describe where the ANOVA and Duncan test were performed.

L201 Why use a ribosomal RNA database to assess nifH gene diversity? The bioinformatic analysis does not reflect the sequencing approach employed in this work. Include in supplementary material the pipeline used or in your github repository.  

The NGS data (raw Illumina reads) should be deposited in an NCBI Bioproject. Please upload that information to NCBI and place this information in the manuscript.

It is also not mentioned if the analyzed variables presented a normal distribution or if normality was evaluated for the appropriate choice of statistical tests.

Results 

When mentioning that one result is lower or higher than another, please indicate with respect to what it was compared.

Write the results indicating a previous description of  experiments to take the reader by the hand throughout the manuscript.

It is mentioned in the text that there were positive and negative relationships between variables, but no statistical analysis of correlation was shown, please include them.

L-244 Please include information on the average number of raw reads obtained from sequencing and the % of them that could be assigned, and to which bacterial genera. Then continue with the diversity analysis.

Why not include a non-plant soil comparison control to see the effect of the cropping system and N fertigation on the native microbiota?

Were the observed differences in microbial groups between treatments supported by statistical analysis? Please mention it in the text. How many replicates of each treatment were performed for nifH gene sequencing by illumina?

This section does not mention how redundancy analysis (RDA) and Mantel test were performed. Please include the eigthvalues of the PCA data shown and the contribution of the analyzed variables to components 1 and 2 shown in Figure 3C.

I recommend that the qPCR data be described after the nifH diversity analysis and then the correlations with soil physicochemical properties.

Discussion

The discussion needs to be improved and include a scientific explanation of the results, not just re-describe them.

Explain why corn favors recruitment of nitrogen fixers to the intercrop.

Discuss the limitation of the primers used, i.e., whether they only target symbiotic or free-living fixers. Many free-living diazotrophs are associated with corn naturally, but did not appear in intercropping treatments. Explain why and contrast with microbes from other intercrops.

How does N concentration affect nif gene expression and nitrogen fixation?

L391 Comment on the limitations of the taxonomic assignment that prevented you from achieving a classification at the family or genus level.

L417, please review the correct way to cite references in the text.

Discuss the relevance of your study to agricultural production and the use of intercropping in the field. 

How can your results be applied to other intercrops?

Conclusion

The conclusion should be much more synthesized and highlight the relevant findings of the research. Give a critical overview of the findings and perspectives of your work.

Figures and tables

Figures and tables should be self-explanatory, please mention in the figure captions the statistical test and the meaning of the letters appearing in the figures. Explain or avoid the use of abbreviations as well.

The figures have very low quality, especially figure 3, please improve that aspect. I suggest separating panels B and C of figure 3 and explaining each figure extensively.

The supplementary tables should be explained in detail and the abbreviations used should be defined.

References 

Review the correct ways to cite references and update with 2023 references.

Author Response

Responses to the comments

Manuscript Number: agronomy-2281747

We thank you and the reviewers for the very helpful comments on our manuscript ‘ Changes in Soil Rhizobia Diversity and their effects on the Symbiotic Efficiency of Soybean Intercropped with Maize(agronomy-2281747)’. We have carefully considered all comments and thoroughly revised the text. The English of the whole text has been modified thoroughly by a native speaker. We also have prepared all of the items required by the submission system. We hope the revised manuscript is suitable for publication in Agronomy.

All our responses to reviewers are detailed in blue. Changes made in the manuscript are highlighted in red. The revision has been developed in consultation with all coauthors, and each author has given approval to the final form of this revision.

Prof. Shumin Li

Resource and Environmental College, Northeast Agricultural University

Suggestions are detailed below:Please check the English with a native reviewer, as many sentences are very long and repetitive.

 Response: The paper has been carefully revised by a professional language editing service to improve the grammar and readability.

Abstract

Please specify nitrogen concentrations after mentioning the treatment. 

Response: Thanks for your suggestion. I have indicated the nitrogen concentration after each treatment(Line13)

 L20-21, it is not understood to which variables the percentages correspond.

Response: Thanks for your suggestion. This sentence has been improved to“The nitrogen fixing efficiency in N0, N1 and N2 treatments increased by 69%, 59%, 42% and nodule number of soybean was  10%, 22%, 21%, respectively, compared with monocultures.”(Line20-22)

L22 Specify with respect to which treatment an increase of nitrogen-fixing bacteria diversity was observed 

Response: Thanks for your suggestion. This sentence has been improved to“The soybean nitrogen-fixing bacteria diversity in intercropping under N0 and N1 treatments significantly increased compared with monocultures.”(Line22-23)

L25 Specify what AN and other abbreviations mean. 

Response: Thanks for your suggestion. The meaning of AN and other abbreviationshas been explained.(Line15,27,76,Line113-115)

L26 Conclusions should be limited to the effects of intercropping and nitrogen supply on rhizobia diversity. 

Response: Thanks for your suggestion. This sentence has been improved to“These results help us to understand the mechanisms that nitrogen use efficiency was improved and nitrogen fertilizer could be reduced in legume/ Gramineae intercropping, which is important to improve the sustainability of agricultural production.”(Line27-30)

Introduction 

Add data from the bean/maize system, where there is already work on microbial diversity and its transcriptomic responses. 

Response: Thanks for your suggestion. We have added data on microbial diversity from maize/soybean intercropping systems to the introduce. See(Line41-43)

Reference

Chen, P., Du, Q., Liu, X., Zhou, L., Hussain, S., Lei, L., Song, C., Wang, X., Liu, W., Yang, F., Shu, K., Liu, J., Du, J., Yang, W., & Yong, T. (2017). Effects of reduced nitrogen inputs on crop yield and nitrogen use efficiency in a long-term maize-soybean relay strip intercropping system. PLoS One, 12(9), e0184503. https://doi.org/10.1371/journal.pone.0184503 Fu, Z.-d., Zhou, L., Chen, P., Du, Q., Pang, T., Song, C., Wang, X.-c., Liu, W.-g., Yang, W.-y., & Yong, T.-w. (2019). Effects of maize-soybean relay intercropping on crop nutrient uptake and soil bacterial community. Journal of integrative agriculture, 18(9), 2006-2018. https://doi.org/10.1016/s2095-3119(18)62114-8

L73, 76 Review the correct way to cite in-text references  

Response: I apologies for this mistake. I have updated the way of reference.(Line80-83)

L85, what do you mean by gene replication? Rephrase the idea please

 Response: Thanks for your suggestion. This sentence has been improved to“Fertilization increased the abundance of nitrogen-fixing bacteria species and was beneficial to increase the number and variety nitrogen-fixing bacterial genes in the black soil area of Northeast China”(Line82-84)

L86, be sure to spell scientific names correctly.

Response: I apologies for this mistake. This sentence has been improved to“Evaluation of nifH gene diversity in the rhizosphere of two sorghums(Sorghum Moench) varieties suggested that nitrogen application was the main factor affecting the nitrogen-fixing bacteria community composition in sorghum”.(Line84-86)

I recommend first explaining aspects of bacterial nitrogenase regulation and then the findings associated with N-fertilization.

Response: Thanks for your suggestion. We explained the nifH gene encoding nitrogen-fixing enzyme in nitrogen-fixing microorganisms, and then explained the effect of nitrogen application on nifH gene.(Line61-95)

Include more references to work on the effect of rhizobial diversity and nitrogen fixation in soybeans in response to high nitrogen levels, either in the introduction or the discussion. 

Response: Thanks for your suggestion. We have updated references to soybean rhizobial diversity and nitrogen fixation at high nitrogen levels in the introduction and discussion.(Line 91,422,424)

Reference

Nguyen, H. P., Miwa, H., Obirih-Opareh, J., Suzaki, T., Yasuda, M., & Okazaki, S. (2020). Novel rhizobia exhibit superior nodulation and biological nitrogen fixation even under high nitrate concentrations. FEMS Microbiol Ecol, 96(2). https://doi.org/10.1093/femsec/fiz184 Zhou, J., Jiang, X., Wei, D., Zhao, B., Ma, M., Chen, S., Cao, F., Shen, D., Guan, D., & Li, J. (2017). Consistent effects of nitrogen fertilization on soil bacterial communities in black soils for two crop seasons in China. Sci Rep, 7(1), 3267. https://doi.org/10.1038/s41598-017-03539-6 Ren, N., Wang, Y., Ye, Y., Zhao, Y., Huang, Y., Fu, W., & Chu, X. (2020). Effects of Continuous Nitrogen Fertilizer Application on the Diversity and Composition of Rhizosphere Soil Bacteria. Front Microbiol, 11, 1948. https://doi.org/10.3389/fmicb.2020.01948

Materials and methods

In the experimental design appeared results of soil properties, those tests were performed by you or are previously reported?  Specify and do not include results in the methodology. 

Response: All soil property results in the test design were determined by ourself ,This is the basic property of the soil before planting .(Line115-117)

Mention the methods used for each chemical determination and its reference. 

Response: Thanks for your suggestion. We have added references and methods for chemical determination.(Line146-153)

L150, is the dry weight of the plant or nodules? Be more specific about which sample was analyzed.

Response: Thanks for your suggestion. It is the dry weight of soybean nodules .This sentence has been improved to“Measurement of Soybean Nodules and Nodules Dry Weight”(Line154 )

162, specify where the DNA was extracted from. Soil, rhizosphere or from bacteria? 

Response: Thanks for your suggestion. the DNA was extracted from rhizosphere soil. (Line171)

L163 Include more information about the DNA extraction, include the brand of the kit and if the manufacturer's recommendations were followed. I could not find it on the web. 

Response: Thanks for your suggestion. We added information such as DNA extraction kit information and manufacturer's opinion(Line166-186)

L165, why were these primers selected, what region do they amplify and for which group of diazotrophs is it effective. What is the size of the PCR product? Were adapters included for the sequencing platform, please include the sequence in supplementary material?

Response: nifH gene is an important structural gene of azofixase in nitrogen-fixing microorganisms, and it is often used to detect a molecular index of nitrogen-fixing bacteria. It is also the gene with the most in-depth study on nif operon, which can obtain a wide range of sequences from multiple environments. The PCR product size 450, nifHF: AAAGGYGGWATCGGYAARTCCACCAC, nifHR: TTGTTSGCSGCRTACATSGCCATCAT. We added this content to the materials and methods. (Line175-178)

L173 Could you please add how you performed the copy number calibration and standard curve? Include how many replicates were done for qPCR and normalization methods.  

Response: The copy number was calibrated using CT=-k lg X0 + b ;Preparation of standard curve: 10 times gradient dilution of each constructed plasmid, 45μl diluent +5μl plasmid, generally do 4-6 points, through the preliminary experiment respectively select 10-3 ~ 10-8 diluent of standard for preparation of standard curve.The number of QPCR was repeated was 3 times, which was normalized by internal reference genes. We added this content to the materials and methods. (Line189-194)

L176, explain in an orderly fashion how 15N was determined, from sample collection to concentration calculations.

Response: After the samples were harvested, δ15N data were obtained by elemental analyser-isotope ratio mass spectrometer, and then calculated. We added this content to the materials and methods. (Line201-202)

L196 Describe where the ANOVA and Duncan test were performed.

Response: Thanks for your suggestion. The statistical analysis was carried out by R language(R, V4.2.1), that is, the difference of different treatments was analyzed by one-way variance analysis (ANOVA), and the significance of the difference by Duncan multiple repetition range test (P<0.05), the difference of Two-ways ANOVA variance analyzed the differences among different factors. We added this content to the materials and methods. (Line223-227)

L201 Why use a ribosomal RNA database to assess nifH gene diversity? The bioinformatic analysis does not reflect the sequencing approach employed in this work. Include in supplementary material the pipeline used or in your github repository.  

Response: I apologies for this mistake. This sentence has been improved to“The bacterial composition and differences between groups were analyzed and compared on the Majorbio I-Sanger cloud platform and the nifH database”(Line230-231)

The NGS data (raw Illumina reads) should be deposited in an NCBI Bioproject. Please upload that information to NCBI and place this information in the manuscript.

Response: Thanks for your suggestion. Purified amplicons were pooled in equimolar amounts and paired-end sequenced on an Illumina MiSeq PE300 platform (Illumina, San Diego,USA) according to the standard protocols by Majorbio Bio-Pharm Technology Co. Ltd. (Shanghai, China). The raw sequencing reads were deposited into the NCBI Sequence Read Archive (SRA) database (Accession Number: PRJNA945238). We added this content to the materials and methods. (Line195-199)

It is also not mentioned if the analyzed variables presented a normal distribution or if normality was evaluated for the appropriate choice of statistical tests.

Response: Thanks for your suggestion. This study mentions that all variables are tested by R language “fBascis package” to see whether they conform to normal distribution. We added this content to the materials and methods. (Line227)

Results 

When mentioning that one result is lower or higher than another, please indicate with respect to what it was compared.

Response: Thanks for your suggestion. We have indicated out the relevant aspects of comparison. We added this content to the result .(Line239-241,248-252,262-264)

Write the results indicating a previous description of  experiments to take the reader by the hand throughout the manuscript.

Response: Thanks for your suggestion. We have added the results described in previous experiments so that readers can walk through them throughout the manuscript. We added this content to the result.(Line245,336,345)

It is mentioned in the text that there were positive and negative relationships between variables, but no statistical analysis of correlation was shown, please include them.

Response: Thanks for your suggestion. We added the Pearson correlation analysis method to materials and methods.(Line229)

L-244 Please include information on the average number of raw reads obtained from sequencing and the % of them that could be assigned, and to which bacterial genera. Then continue with the diversity analysis.

Response: Thanks for your suggestion. A total of 115396 reads obtained from sequencing and the 66.22% of them that could be assigned. We added this content to the result. (Line289-290)

Why not include a non-plant soil comparison control to see the effect of the cropping system and N fertigation on the native microbiota?

Response: Thanks for your suggestion. Our aim was to compare the variation of fixation bacteria diversity and abundance in rhizosphere soil of soybean after intercropping, so we used monoculture soybean as a control, without setting no-plant soil as comparison control.

Were the observed differences in microbial groups between treatments supported by statistical analysis? Please mention it in the text. How many replicates of each treatment were performed for nifH gene sequencing by illumina?

Response: Thanks for your suggestion. Statistical analysis supported microbial differences observed between treatments, with three replicates per treatment of nifH gene determined by illumina. We added this content to the materials and methods. (Line189-190)

This section does not mention how redundancy analysis (RDA) and Mantel test were performed. Please include the eigenvalues of the PCA data shown and the contribution of the analyzed variables to components 1 and 2 shown in Figure 3C.

Response: Thanks for your suggestion. We added the analysis methods of RDA and mantel test to the materials and methods.(Line228-229)We include the eigenvalues of the data and the variables' contributions to component1 and 2 in the results.(Line339-343)

I recommend that the qPCR data be described after the nifH diversity analysis and then the correlations with soil physicochemical properties.

Response: Thanks for your suggestion. We first analyzed the diversity and gene abundance of nitrogen-fixing microorganisms, and then analyzed their physical and chemical properties with soil.(Figure3 and Figure4)

Discussion

The discussion needs to be improved and include a scientific explanation of the results, not just re-describe them.

Response: Thanks for your suggestion. We refined the discussion and scientifically explained the results. We added this content to the discuss.(Line373-374,381-382,387-391,479-481)

Explain why corn favors recruitment of nitrogen fixers to the intercrop.

Response: Soil available N was decreased uue to the competition of maize roots in intercropping, Which lead to nitrogen fixers improved. We added this content to the discuss.(Line373-374)

Discuss the limitation of the primers used, i.e., whether they only target symbiotic or free-living fixers. Many free-living diazotrophs are associated with corn naturally but did not appear in intercropping treatments. Explain why and contrast with microbes from other intercrops. 

Response: The nifH gene is the common that related to nitrogen fixation gene of bean. It can reflect the nitrogen fixation ability of bean.

How does N concentration affect nifH gene expression and nitrogen fixation?

 Response: We added this content to the discussion.(461-467,479-481)

L391 Comment on the limitations of the taxonomic assignment that prevented you from achieving a classification at the family or genus level.

Response: Thanks for your suggestion. We amend the unclassified_k__norank_d__Bacteria and unclassified_p_Proteobacteria to g_unclassified_p_Proteobacteria and g_unclassified_k__norank_d__Bacteria. To indicate that they belong to some unclassified genus of bacteria.(Line429,432)

L417, please review the correct way to cite references in the text. 

Response: I apologies for this mistake. We examined the method of citing references.(Line459)

Discuss the relevance of your study to agricultural production and the use of intercropping in the field. 

 Response: Maize/soybean intercropping has been widely used to improve crop and feed yields, grain nutrient improvement, and soil health. It is important to explore the optimum nitrogen supply of intercropping soybean to improve crop yield.

How can your results be applied to other intercrops? 

Response: Results of the experiment can provide the opinion how to improve the nitrogen fixing efficiency in legumes/ Gramineae intercropping system and reduce chemical fertilizer application.

Conclusion 

The conclusion should be much more synthesized and highlight the relevant findings of the research. Give a critical overview of the findings and perspectives of your work.

 Response: Thanks for your suggestion. We have a new overview of the summary(Line 485-494)

Figures and tables

 Figures and tables should be self-explanatory, please mention in the figure captions the statistical test and the meaning of the letters appearing in the figures. Explain or avoid the use of abbreviations as well. 

Response: Thanks for your suggestion. We have indicated the statistical test and the meaning of the letters in the figures and tables and explained or avoided the use of abbreviations.(Figure1,2,3,4,5)

The figures have very low quality, especially figure 3, please improve that aspect. I suggest separating panels B and C of figure 3 and explaining each figure extensively.

Response: Thanks for your suggestion. We improved the picture quality and separated b and c in Figure 3 and explained them in detail.(Figure3b and Figure4a)

The supplementary tables should be explained in detail and the abbreviations used should be defined.

Response: Thanks for your suggestion. We have a detailed explanation of the supplementary form and a detailed explanation of abbreviations.(Table 2,3)

References Review the correct ways to cite references and update with 2023 references.

Response: Thanks for your suggestion. We examined the method of citing references and updated the references for 2023. See reference[37],[51],[54]

The revised manuscript is attached

Round 2

Reviewer 1 Report

Manuscript ID: agronomy-2281747

Title: Change of Soil Rhizobia Diversity and its effects on the Symbiotic Efficiency of Soybean Intercropped with Maize

Authors: Zeyu Cheng , Lingbo Meng , Tengjiao Yin , Ying Li , Yuhang Zhang , Shumin Li *

General Comments

The authors have appropriately responded the reviewer’s comments and revised the manuscript. The following miner points need to be revised.

Line150: Isn't “alkaline nitrogen” a misspelling of “available nitrogen” ?

Line175-8: The reference of nifH-F and mifH-R is required.

Figure 5a and 5b: “nifH” in the label of the vertical axis should be italicized. The unit notation in Figure 5a should be correct. (“g /soil” is not correct.)

Table 1: The unit notation (no·plant and g·plant) should be correct (No. plant-1 and g plant-1?).

Line 279: It looks like there is an unwanted additional space between “supply level” and “, and different”.

Figure 3: Since the figures show bacterial community composition, the title is not appropriate.

Figure 3a: Genera (Bradyrhizobium, Skemanella, and Azohydromonas) should be italicized.

Line 394-5: It is unclear what “fertilizer efficiency” indicates. Because the authors did not seem to discuss it, “and fertilizer” should be deleted in this sentence.

Line 395-9: Because the effect of nitrogen application rate on the soybean biomass did not reach a significant level as the authors mentioned (Line 259-60, Figure 1d), they cannot conclude that the biomass was lower than that of N1 treatment. Therefore, these sentences (Line 395-9) should be revised.

Figure 4b: The authors should explain figure 4b more in its figure legend.

Figure 5: “(a)” should be written between “level (x107)” and “and their”. The explanation of Figure 5b should be correct because this figure contains the plots of soybean monocropping (SS).

Line 479: Since this research did not examine the nitrogenase and its activity, the authors cannot describe that appropriate nitrogen application can activate nitrogenase.

Figure legends: The authors should state what the bars indicate. (Standard deviation of 5 samples?)

Regarding the unit notation, both “/” and “XX-1” are presented in this manuscript. The unit notation should be unified in the manuscript including tables and figures. (For example, “copies/g soil” in Line 358 and “kg N ha-1” in Line 379.)

Regarding following comment and response, my previous comments did not seem to have been understood by the authors correctly, so I commented again.

Statements that imply that soybean fixes nitrogen should be avoided (Line 71, 208, and 215 etc...).

Response: Thanks for your suggestion. We avoid using statements that imply that soybean fixes nitrogen(Line79,233,240 etc...)

I think that "the nitrogen fixation efficiency of soybean" (Line 77-8, 238, and 245) and “the nitrogen fixed by soybeans” (Line 219-20) are inaccurate because nitrogen is fixed by nitrogen fixing bacteria, not soybean. Since the bacteria fix nitrogen in nodule of soybean, it could be described as "nitrogen fixation of (by) soybean," but it would be better to describe it so as not to mislead readers.

I hope these comments will be helpful in completing this paper.

Author Response

Dear Reviewer
Thank you for your  help with our manuscript "agronomy-2281747".

Reviewer 3 Report

Dear authors,

A few last comments before acceptance. 

In methodology, it was mentioned that by qPCR, the abundance of AOB, Nitrospira, Nitrobacter and total bacteria was measured, however, previously it had only been used to measure nifH levels in the rhizosphere. Why are new methodologies appearing now and where are the results of AOB, nitrospira and total bacteria quantification?

Please include a table with the sequences of primers used in the qPCR and the reference of each one. nifH, AOB, Nitrospira, Nitrobacter and total bacteria

Why quantify anammox bacteria now?

In both methodologies and figures, please mention the number of replicates per treatment or experiment. If asterisks appear in the figures, mention the p-value and the statistical test used. Be more detailed in figure captions 3 and 4.

Please separate quantitative real-time PCR (qPCR) and high-throughput sequencing data analysis ( section Determination of Nitrogen-fixing Bacteria Diversity).

In the methodology. Please add a section on bioinformatics analysis, indicating the preprocessing of the sequencing reads and the pipeline of analysis of the nifH sequences. So that it can be reproducible by the scientific community. Separate it from statistical analyses.

Please add a reference or link to the nifH database used

Is it known which species of bradyrhizobia are nodulating the plants, and is it similar between treatments?

Please describe at the species level the abundances of nifH genes associated with Bradyrhizobium.

Best,

Author Response

(The authors gave the same response as above.)
